# Learning Neurosymbolic Generative Models via Program Synthesis

## Abstract

Significant strides have been made toward designing better generative models in recent years. Despite this progress, however, state-of-the-art approaches are still largely unable to capture complex global structure in data. For example, images of buildings typically contain spatial patterns such as windows repeating at regular intervals; state-of-the-art generative methods can't easily reproduce these structures. We propose to address this problem by incorporating programs representing global structure into the generative model—e.g., a 2D for-loop may represent a configuration of windows. Furthermore, we propose a framework for learning these models by leveraging program synthesis to generate training data. On both synthetic and real-world data, we demonstrate that our approach is substantially better than the state-of-the-art at both generating and completing images that contain global structure.

## 1 Introduction

There has been much interest recently in generative models, following the introduction of both variational autoencoders (VAEs) Kingma & Welling (2014) and generative adversarial networks (GANs) Goodfellow et al. (2014). These models have successfully been applied to a range of tasks, including image generation Radford et al. (2015), image completion IIzuka et al. (2017), texture synthesis Jetchev et al. (2017); Xian et al. (2018), sketch generation Ha & Eck (2017), and music generation Dieleman et al. (2018).

Despite their successes, generative models still have difficulty capturing global structure. For example, consider the image completion task in Figure 1. The original image (left) is of a building, for which the global structure is a 2D repeating pattern of windows. Given a partial image (middle left), the goal is to predict the completion of the image. As can be seen, a state-of-the-art image completion algorithm has trouble reconstructing the original image (right) IIzuka et al. (2017). [1] Real-world data often contains such global structure, including repetitions, reflectional or rotational symmetry, or even more complex patterns.

In the past few years, *program synthesis* Solar-Lezama et al. (2006) has emerged as a promising approach to capturing patterns in data Ellis et al. (2015; 2018); Valkov et al. (2018). The idea is that simple programs can capture global structure that evades state-of-the-art deep neural networks. A key benefit of using program synthesis is that we can design the space of programs to capture different kinds of structure—e.g., repeating patterns Ellis et al. (2018), symmetries, or spatial structure Deng et al. (2018)—depending on the application domain. The challenge is that for the most part, existing approaches have synthesized programs that operate directly over raw data. Since programs have difficulty operating over perceptual data, existing approaches have largely been limited to very simple data—e.g., detecting 2D repeating patterns of simple shapes Ellis et al. (2018).

We propose to address these shortcomings by synthesizing programs that represent the underlying structure of high-dimensional data. In particular, we decompose programs into two parts: (i) a *sketch* $s \in S$ that represents the skeletal structure of the program Solar-Lezama et al. (2006), with *holes* that are left unimplemented, and (ii) *components* $c \in C$ that can be used to fill these holes. We consider *perceptual components*—i.e., holes in the sketch are filled with raw perceptual data. For example, the

---

[1] The baseline model performs particularly poorly since our dataset is small. We show on synthetic data that our approach is significantly better even when a large amount of data is available.

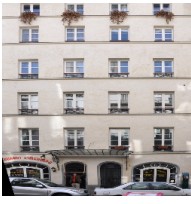 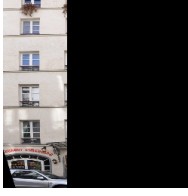 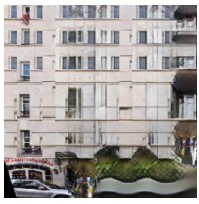 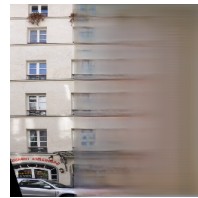

original image $x^*$    partial image $x_{\text{part}}$    completion $\hat{x}$ (ours)    completion $\hat{x}$ (baseline)

Figure 1: The task is to complete the partial image $x_{\text{part}}$ (middle left) into an image that is close to the original image $x^*$ (left). By incorporating programmatic structure into generative models, the completion (middle right) is able to substantially outperform a state-of-the-art baseline IIzuka et al. (2017) (right). Note that not all non-zero pixels in the sketch rendering retain the same value in the completed picture due to the nature of the following completion process

program

```
for i = 1..3
    for j = 1..4
        draw(i*2, j*3,      )
```

represents the structure in the original image $x^*$ in Figure 1 (left). The black text is the sketch, and the component is a sub-image taken from the given partial image. Then, the `draw` function renders the given sub-image at the given position. We call a sketch whose holes are filled with perceptual components a *neurosymbolic program*.

Building on these ideas, we propose an approach called *program-synthesis (guided) generative models* (PS-GM) that combines neurosymbolic programs representing global structure with state-of-the-art deep generative models. By incorporating programmatic structure, PS-GM substantially improves the quality of these state-of-the-art models. As can be seen, the completion produced using PS-GM (middle right of Figure 1) substantially outperforms the baseline.

We show that PS-GM can be used for both generation from scratch and for image completion. The generation pipeline is shown in Figure 2. At a high level, PS-GM for generation operates in two phases:

- First, it generates a program that represents the global structure in the image to be generated. In particular, it generates a program $P = (s, c)$ representing the latent global structure in the image (left in Figure 2), where $s$ is a sketch (in the domain considered here, a list of 12 for-loop structures) and $c$ is a perceptual component (in the domain considered here, a list of 12 sub-images).
- Second, our algorithm executes $P$ to obtain a *structure rendering* $x_{\text{struct}}$ representing the program as an image (middle of Figure 2). Then, our algorithm uses a deep generative model to complete $x_{\text{struct}}$ into a full image (right of Figure 2). The structure in $x_{\text{struct}}$ helps guide the deep generative model towards images that preserve the global structure.

The image-completion pipeline (see Figure 3) is similar.

Training these models end-to-end is challenging, since a priori, ground truth global structure is unavailable. Furthermore, representative global structure is very sparse, so approaches such as reinforcement learning do not scale. Instead, we leverage domain-specific program synthesis algorithms to produce examples of programs that represent global structure of the training data. In particular, we propose a synthesis algorithm tailored to the image domain, which extracts programs with nested for-loops that can represent multiple 2D repeating patterns in images. Then, we use these example programs as supervised training data.

Our programs can capture rich spatial structure in the training data. For example, in Figure 2, the program structure encodes a repeating structure of 0's and 2's on the whole image, and a separate repeating structure of 3's on the right-hand side of the image. Furthermore, in Figure 1, the generated image captures the idea that the repeating pattern of windows does not extend to the bottom portion of the image.

```
for i = 1..3
  for j = 1..4
    draw(i*2 + 2, j*2 + 1, [] )
```

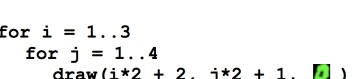
for loop from sampled program $P$

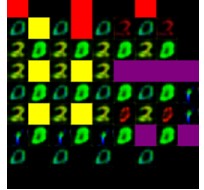
structure rendering $x_{\text{struct}}$

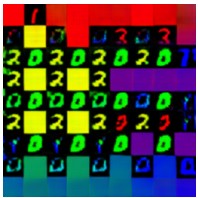
completed image $x$

Figure 2: Our image generation pipeline consists of the following steps: (i) Our generative model samples a latent vector $z \sim p(z)$, and samples a program $P = (s, c) \sim p_\phi(s, c \mid z)$ (one loop of which is shown left). (ii) Our model executes $P$ to obtain a rendering of the program structure $x_{\text{struct}}$ (middle). (iii) Our model samples a completion $x \sim p_\theta(x \mid s, c)$ of $x_{\text{struct}}$ into a full image (right).

**Contributions.** We propose an architecture of generative models that incorporates programmatic structure, as well as an algorithm for training these models (Section 2). Our learning algorithm depends on a domain-specific program synthesis algorithm for extracting global structure from the training data; we propose such an algorithm for the image domain (Section 3). Finally, we evaluate our approach on synthetic data and on a real-world dataset of building facades Tyleček & Šára (2013), both on the task of generation from scratch and on generation from a partial image. We show that our approach substantially outperforms several state-of-the-art deep generative models (Section 4).

**Related work.** There has been growing interest in applying program synthesis to machine learning, for purposes of interpretability Wang & Rudin (2015); Verma et al. (2018), safety Bastani et al. (2018), and lifelong learning Valkov et al. (2018). Most relevantly, there has been interest in using programs to capture structure that deep learning models have difficulty representing Lake et al. (2015); Ellis et al. (2015; 2018); Pu et al. (2018). For instance, Ellis et al. (2015) proposes an unsupervised learning algorithm for capturing repeating patterns in simple line drawings; however, not only are their domains simple, but they can only handle a very small amount of noise. Similarly, Ellis et al. (2018) captures 2D repeating patterns of simple circles and polygons; however, rather than synthesizing programs with perceptual components, they learn a simple mapping from images to symbols as a preprocessing step. The closest work we are aware of is Valkov et al. (2018), which synthesizes programs with *neural components* (i.e., components implemented as neural networks); however, their application is to lifelong learning, not generation, and to learning with supervision (labels) rather than to unsupervised learning of structure.

Additionally, there has been work extending neural module networks Andreas et al. (2016) to generative models Deng et al. (2018). These algorithms essentially learn a collection of neural components that can be composed together based on hierarchical structure. However, they require that the structure be available (albeit in natural language form) both for training the model and for generating new images.

Finally, there has been work incorporating spatial structure into generative models for generating textures Jetchev et al. (2017); however, their work only handles a single infinite repeating 2D pattern. In contrast, we can capture a rich variety of spatial patterns parameterized by a space of programs. For example, the image in Figure 1 generated by our technique contains different repeating patterns in different parts of the image.

## 2 GENERATIVE MODELS WITH LATENT STRUCTURE

We describe our proposed architecture for generative models that incorporate programmatic structure. For most of this section, we focus on generation; we discuss how we adapt these techniques to image completion at the end. We illustrate our generation pipeline in Figure 2.

Let $p_{\theta,\phi}(x)$ be a distribution over a space $\mathcal{X}$ with unknown parameters $\theta, \phi$ that we want to estimate. We study the setting where $x$ is generated based on some latent structure, which consists of a *program sketch* $s \in \mathcal{S}$ and a *perceptual component* $c \in \mathcal{C}$, and where the structure is in turn generated

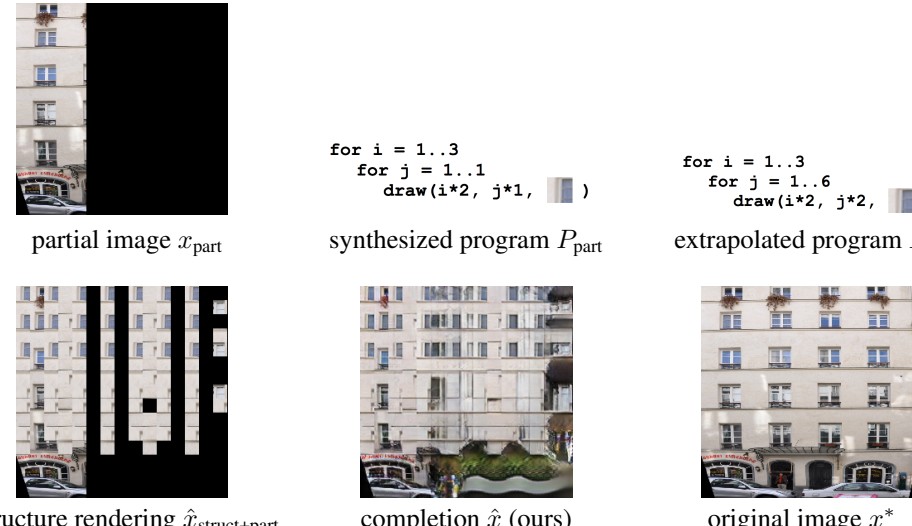

Figure 3: Our image completion pipeline consists of the following steps: (i) Given a partial image $x_{\text{part}}$ (top left), our program synthesis algorithm (Section 3) synthesizes a program $P_{\text{part}}$ representing the structure in the partial image (top middle). (ii) Our model $f$ extrapolates $P_{\text{part}}$ to a program $\hat{P} = f(P_{\text{part}})$ representing the structure of the whole image. (iii) Our model executes $\hat{P}$ to obtain a rendering of the program structure $\hat{x}_{\text{struct+part}}$ (bottom left). (iv) Our model completes $\hat{x}_{\text{struct+part}}$ into an image $\hat{x}$ (bottom middle), which resembles the original image $x^*$ (bottom right).

conditioned on a latent vector $z \in \mathcal{Z}$—i.e.,

$$p_{\theta,\phi}(x) = \int_{\mathcal{Z}} \int_{\mathcal{C}} \sum_{s \in \mathcal{S}} p_\theta(x \mid s, c) p_\phi(s, c \mid z) p(z) dc dz.$$

Figure 2 shows a single for loop out of the twelve for loops determined by a sampled program $P = (s, c) \sim p_\phi(s, c \mid z)$ (left), and an example of the sampled completion $x \sim p_\theta(x \mid s, c)$ (right). To sample a completion, our model executes $P$ to obtain a structure rendering $x_{\text{struct}} = \text{eval}(P)$ (middle), and then samples a completion based on $x_{\text{struct}}$—i.e., $p_\theta(x \mid s, c) = p_\theta(x \mid x_{\text{struct}})$.

We now describe our algorithm for learning the parameters $\theta, \phi$ of $p_{\theta,\phi}$, followed by a description of our choices of architecture for $p_\phi(s, c \mid z)$ and $p_\theta(x \mid s, c)$.

**Learning algorithm.** Given training data $\{x^{(i)}\}_{i=1}^n \subseteq \mathcal{X}$, where $x^{(i)} \sim p_{\theta,\phi}(x)$, the maximum likelihood estimate is

$$\theta^*_{\text{MLE}}, \phi^*_{\text{MLE}} = \arg\max_{\theta,\phi} \sum_{i=1}^n \log p_{\theta,\phi}(x^{(i)}).$$

Since $\log p_{\theta,\phi}(x)$ is intractable to optimize, we use an approach based on the variational autoencoder (VAE). In particular, we use a variational distribution

$$q_{\tilde{\phi}}(s, c, z \mid x) = q_{\tilde{\phi}}(z \mid s, c) q(s, c \mid x),$$

which has parameters $\tilde{\phi}$. Then, we optimize $\tilde{\phi}$ while simultaneously optimizing $\theta, \phi$. Using $q_{\tilde{\phi}}(s, c, z \mid x)$, the *evidence lower bound* on the log-likelihood is

$$\log p_{\theta,\phi}(x) \geq \mathbb{E}_{q(s,c,z|x)}[\log p_\theta(x \mid s, c)] - D_{\text{KL}}(q(s, c, z \mid x) \parallel p_\phi(s, c \mid z)p(z)) \qquad (1)$$
$$= \mathbb{E}_{q(s,c|x)}[\log p_\theta(x \mid s, c)] + \mathbb{E}_{q(s,c|x), q_{\tilde{\phi}}(z|s,c)}[\log p_\phi(s, c \mid z)]$$
$$- \mathbb{E}_{q(s,c|x)}[D_{\text{KL}}(q_{\tilde{\phi}}(z \mid s, c) \parallel p(z))] - H(q(s, c \mid x)),$$

where $D_{\text{KL}}$ is the KL divergence and $H$ is information entropy. Thus, we can approximate $\theta^*, \phi^*$ by optimizing the lower bound (1) instead of $\log p_{\theta,\phi}(x)$. However, (1) remains intractable since we are integrating over all program sketches $s \in \mathcal{S}$ and perceptual components $c \in \mathcal{C}$. Using sampling to estimate these integrals would be very computationally expensive. Instead, we propose an approach that uses a single point estimate of $s_x \in \mathcal{S}$ and $c_x \in \mathcal{C}$ for each $x \in \mathcal{X}$, which we describe below.

**Synthesizing structure.** For a given $x \in \mathcal{X}$, we use *program synthesis* to infer a *single* likely choice $s_x \in \mathcal{S}$ and $c_x \in \mathcal{C}$ of the latent structure. The program synthesis algorithm must be tailored to a specific domain; we propose an algorithm for inferring for-loop structure in images in Section 3. Then, we use these point estimates in place of the integrals over $\mathcal{S}$ and $\mathcal{C}$—i.e., we assume that

$$q(s, c \mid x) = \delta(s - s_x)\delta(c - c_x),$$

where $\delta$ is the Dirac delta function. Plugging into (1) gives

$$\log p_{\theta,\phi}(x) \geq \log p_\theta(x \mid s_x, c_x) + \mathbb{E}_{q_{\tilde{\phi}}(z \mid s_x, c_x)}[\log p_\phi(s_x, c_x \mid z)] - D_{\mathrm{KL}}(q_{\tilde{\phi}}(z \mid s_x, c_x) \parallel p(z)). \tag{2}$$

where we have dropped the degenerate terms $\log \delta(s - s_x)$ and $\log \delta(c - c_x)$ (which are constant with respect to the parameters $\theta, \phi, \tilde{\phi}$). As a consequence, (1) decomposes into two parts that can be straightforwardly optimized—i.e.,

$$\begin{aligned} \log p_{\theta,\phi}(x) \geq\ & \mathcal{L}(\theta; x) + \mathcal{L}(\phi, \tilde{\phi}; x) \\ \mathcal{L}(\theta; x) =\ & \log p_\theta(x \mid s_x, c_x) \\ \mathcal{L}(\phi, \tilde{\phi}; x) =\ & \mathbb{E}_{q_{\tilde{\phi}}(z \mid s_x, c_x)}[\log p_\phi(s_x, c_x \mid z)] - D_{\mathrm{KL}}(q_{\tilde{\phi}}(z \mid s_x, c_x) \parallel p(z)), \end{aligned}$$

where we can optimize $\theta$ and $\phi, \tilde{\phi}$ independently:

$$\theta^* = \arg\max_\theta \sum_{i=1}^n \mathcal{L}(\theta; x^{(i)}), \qquad \phi^*, \tilde{\phi}^* = \arg\max_{\phi, \tilde{\phi}} \sum_{i=1}^n \mathcal{L}(\phi, \tilde{\phi}; x^{(i)}).$$

**Latent structure VAE.** Note that $\mathcal{L}(\phi, \tilde{\phi}; x)$ is exactly equal to the objective of a VAE, where $q_{\tilde{\phi}}(z \mid s, c)$ is the encoder and $p_\phi(s, c \mid z)$ is the decoder—i.e., learning the distribution over latent structure is equivalent to learning the parameters of a VAE. The architecture of this VAE depends on the representation of $s$ and $c$. In the case of for-loop structure in images, we use a sequence-to-sequence VAE.

**Generating data with structure.** The term $\mathcal{L}(\theta; x)$ corresponds to learning a probability distribution (conditioned on the latent structure $s$ and $c$)—e.g., we can estimate this distribution using another VAE. As before, the architecture of this VAE depends on the representation of $s$ and $c$. Rather than directly predicting $x$ based on $s$ and $c$, we can leverage the program structure more directly by first executing the program $P = (s, c)$ to obtain its output $x_{\mathrm{struct}} = \mathrm{eval}(P)$, which we call a *structure rendering*. In particular, $x_{\mathrm{struct}}$ is a more direct representation of the global structure represented by $P$, so it is often more suitable to use as input to a neural network. The middle of Figure 2 shows an example of a structure rendering for the program on the left. Then, we can train a model $p_\theta(x \mid s, c) = p_\theta(x \mid x_{\mathrm{struct}})$.

In the case of images, we use a VAE with convolutional layers for the encoder $q_\phi$ and transpose convolutional layers for the decoder $p_\theta$. Furthermore, instead of estimating the entire distribution $p_\theta(x \mid s, c)$, we also consider two non-probabilistic approaches that directly predict $x$ from $x_{\mathrm{struct}}$, which is an image completion problem. We can solve this problem using GLCIC, a state-of-the-art image completion model IIzuka et al. (2017). We can also use CycleGAN Zhu et al. (2017), which solves the more general problem of mapping a training set of structured renderings $\{x_{\mathrm{struct}}\}$ to a training set of completed images $\{x\}$. [2]

**Image completion.** In image completion, we are given a set of training pairs $(x_{\mathrm{part}}, x^*)$, and the goal is to learn a model that predicts the complete image $x^*$ given a partial image $x_{\mathrm{part}}$. Compared to generation, our likelihood is now conditioned on $x_{\mathrm{part}}$—i.e., $p_{\theta,\phi}(x \mid x_{\mathrm{part}})$. Now, we describe how we modify each of our two models $p_\theta(x \mid s, c)$ and $p_\phi(s, c \mid z)$ to incorporate this extra information.

First, the programmatic structure is no longer fully latent, since we can observe partial programmatic structure in $x_{\mathrm{part}}$. In particular, we can leverage our program synthesis algorithm to help perform completion. We first synthesize programs $P^*$ and $P_{\mathrm{part}}$ representing the global structure in $x^*$ and $x_{\mathrm{part}}$,

---

[2] Pix2Pix Isola et al. (2017) may seem more appropriate since it takes training pairs $(x_{\mathrm{struct}}, x)$, but CycleGAN outperformed it.

respectively. Then, we can train a model $f$ that predicts $P^*$ given $P_{\text{part}}$—i.e., it extrapolates $P_{\text{part}}$ to a program $\hat{P} = f(P_{\text{part}})$ representing the structure of the whole image. Thus, unlike generation, where we sample a program $\hat{P} = (s, c) \sim p_\phi(s, c \mid z)$, we use the extrapolated program $\hat{P} = f(P_{\text{part}})$.

The second model $p_\theta(x \mid s, c)$ for the most part remains the same, except when we execute $\hat{P} = (s, c)$ to obtain a structure rendering $x_{\text{struct}}$, we render onto the partial image $x_{\text{part}}$ instead of onto a blank image to obtain the final rendering $x_{\text{struct+part}}$. Then, we complete the structure rendering $x_{\text{struct+part}}$ into a prediction of the full image $\hat{x}$ as before (i.e., using a VAE, GLCIC, or CycleGAN).

Our image completion pipeline is shown in Figure 3, including the given partial image (top left), the program $P_{\text{part}}$ synthesized from the partial image (top middle), the extrapolated program $\hat{P}$ (top right), the structure rendering $x_{\text{struct+part}}$ (bottom left), and the predicted full image $\hat{x}$ (bottom middle).

## 3   SYNTHESIZING PROGRAMMATIC STRUCTURE

**Image representation.**   Since the images we work with are very high dimensional, for tractability, we assume that each image $x \in \mathbb{R}^{NM \times NM}$ is divided into a grid containing $N$ rows and $N$ columns, where each grid cell has size $M \times M$ pixels (where $M \in \mathbb{N}$ is a hyperparameter). For example, this grid structure is apparent in Figure 3 (top right), where $N = 15$, $M = 17$ and $N = 9$, $M = 16$ for the facade and synthetic datasets respectively. For $t, u \in [N] = \{1, ..., N\}$, we let $x_{tu} \in \mathbb{R}^{M \times M}$ denote the sub-image at the $(t, u)$ position in the $N \times N$ grid.

**Program grammar.**   Given this structure, we consider programs that draw 2D repeating patterns of $M \times M$ sub-images on the grid. More precisely, we consider programs

$$P = ((s_1, c_1), ..., (s_k, c_k)) \in (S \times C)^k$$

that are length $k$ lists of pairs consisting of a sketch $s \in S$ and a perceptual component $c \in C$; here, $k \in \mathbb{N}$ is a hyperparameter. [3] A sketch $s \in S$ has form

$$s = \textbf{for } (i, j) \in \{1, ..., n\} \times \{1, ..., n'\} \textbf{ do}$$
$$\text{draw}(a \cdot i + b, \ a' \cdot j + b', \ \textbf{??})$$
$$\textbf{end for}$$

where $n, a, b, n', a', b' \in \mathbb{N}$ are undetermined parameters that must satisfy $a \cdot n + b \leq N$ and $a' \cdot n' + b' \leq N$, and where **??** is a hole to be filled by a perceptual component, which is an $M \times M$ sub-image $c \in \mathbb{R}^{M \times M}$. [4] Then, upon executing the $(i, j)$ iteration of the for-loop, the program renders sub-image $I$ at position $(t, u) = (a \cdot i + b, a' \cdot j + b')$ in the $N \times N$ grid. Figure 3 (top middle) shows an example of a sketch $s$ where its hole is filled with a sub-image $c$, and Figure 3 (bottom left) shows the image rendered upon executing $P = (s, c)$. Figure 2 shows another such example.

**Program synthesis problem.**   Given a training image $x \in \mathbb{R}^{NM \times NM}$, our program synthesis algorithm outputs the parameters $n_h, a_h, b_h, n'_h, a'_h, b'_h$ of each sketch $s_h$ in the program (for $h \in [k]$), along with a perceptual component $c_h$ to fill the hole in sketch $s_h$. Together, these parameters define a program $P = ((s_1, c_1), ..., (s_k, c_k))$.

The goal is to synthesize a program that faithfully represents the global structure in $x$. We capture this structure using a boolean tensor $B^{(x)} \in \{0, 1\}^{N \times N \times N \times N}$, where

$$B^{(x)}_{t,u,t',u'} = \begin{cases} 1 & \text{if } d(x_{tu}, x_{t'u'}) \leq \epsilon \\ 0 & \text{otherwise,} \end{cases}$$

where $\epsilon \in \mathbb{R}_+$ is a hyperparameter, and $d(I, I')$ is a distance metric between on the space of sub-images. In our implementation, we use a weighted sum of earthmover's distance between the color histograms of $I$ and $I'$, and the number of SIFT correspondences between $I$ and $I'$.

---

[3]So far, we have assumed that a program is a single pair $P = (s, c)$, but the generalization to a list of pairs is straightforward.

[4]For colored images, we have $I \in \mathbb{R}^{M \times M \times 3}$.

---

**Algorithm 1** Synthesizes a program $P$ representing the global structure of a given image $x \in \mathbb{R}^{NM \times NM}$.

---

   **Input:** $X = \{x\} \subseteq \mathbb{R}^{NM \times NM}$
   $\hat{C} \leftarrow \{x_{tu} \mid t, u \in [N]\}$
   $P \leftarrow \varnothing$
   **for** $h \in \{1, ..., k\}$ **do**
      $s_h, c_h = \arg\max_{(s,c) \in S \times \hat{C}} \ell(P_{h-1} \cup \{(s,c)\}; x)$
      $P \leftarrow P \cup \{(s_h, c_h)\}$
   **end for**
   **Output:** $P$

---

Additionally, we associate a boolean tensor with a given program $P = ((s_1, c_1), ..., (s_k, c_k))$. First, for a sketch $s \in S$ with parameters $a, b, n, a', b', n'$, we define

$$\text{cover}(s) = \{(a \cdot i + b, a' \cdot j + b') \mid i \in [n], j \in [n']\},$$

i.e., the set of grid cells where sketch renders a sub-image upon execution. Then, we have

$$B^{(s)}_{t,u,t',u'} = \begin{cases} 1 & \text{if } (t,u), (t',u') \in \text{cover}(s) \\ 0 & \text{otherwise,} \end{cases}$$

i.e., $B^{(s)}_{t,u,t',u'}$ indicates whether the sketch $s$ renders a sub-image at both of the grid cells $(t, u)$ and $(t', u')$. Then,

$$B^{(P)} = B^{(s_1)} \vee ... \vee B^{(s_k)},$$

where the disjunction of boolean tensors is defined elementwise. Intuitively, $B^{(P)}$ identifies the set of pairs of grid cells $(t, u)$ and $(t', u')$ that are equal in the image rendered upon executing each pair $(s, c)$ in $P$. [5]

Finally, our program synthesis algorithm aims to solve the following optimization problem:

$$P^* = \arg\max_P \ell(P; x) \tag{3}$$

$$\ell(P; x) = \|B^{(x)} \wedge B^{(P)}\|_1 + \lambda\|\neg B^{(x)} \wedge \neg B^{(P)}\|_1,$$

where $\wedge$ and $\neg$ are applied elementwise, and $\lambda \in \mathbb{R}_+$ is a hyperparameter. In other words, the objective of (3) is the number of true positives (i.e., entries where $B^{(P)} = B^{(x)} = 1$), and the number of false negatives (i.e., entries where $B^{(P)} = B^{(x)} = 0$), and computes their weighted sum. Thus, the objective of (3) measures for how well $P$ represents the global structure of $x$.

For tractability, we restrict the search space in (3) to programs of the form

$$P = ((s_1, c_1), ..., (s_k, c_k)) \in (S \times \hat{C})^k$$

$$\hat{C} = \{x_{tu} \mid t, u \in [N]\}.$$

In other words, rather than searching over all possible sub-images $c \in \mathbb{R}^{M \times M}$, we only search over the sub-images that actually occur in the training image $x$. This may lead to a slightly sub-optimal solution, for example, in cases where the optimal sub-image to be rendered is in fact an interpolation between two similar but distinct sub-images in the training image. However, we found that in practice this simplifying assumption still produced viable results.

**Program synthesis algorithm.** Exactly optimizing (3) is in general an NP-complete problem. Thus, our program synthesis algorithm uses a partially greedy heuristic. In particular, we initialize the program to $P = \varnothing$. Then, on each iteration, we enumerate all pairs $(s, c) \in S \times \hat{C}$ and determine the pair $(s_h, c_h)$ that most increases the objective in (3), where $\hat{C}$ is the set of all sub-images $x_{tu}$ for $t, u \in [N]$. Finally, we add $(s_h, c_h)$ to $P$. We show the full algorithm in Algorithm 1.

---

[5]Note that the covers of different sketches in $P$ can overlap; we find that ignoring this overlap does not significantly impact our results.

| Model | Score |
|---|---|
| PS-GM (CycleGAN) | **85.51** |
| BL (SpatialGAN) | 258.68 |
| PS-GM (VED) | **59414.7** |
| BL (VAE) | 60368.4 |
| PS-GM (VED Stage 1 $p_\phi(s, c \mid z)$) | 32.0 |
| PS-GM (VED Stage 2 $p_\theta(x \mid s, c)$) | 59382.6 |

Table 1: Performance of our approach PS-GM versus the baseline (BL) for generation from scratch. We report Fréchet inception distance for GAN-based models, and negative log-likelihood for the VAE-based models

| Model | Synthetic | | Facades | |
|---|---|---|---|---|
| | PS-GM | BL | PS-GM | BL |
| GLCIC | **106.8** | 163.66 | **141.8** | 195.9 |
| CycleGAN | **91.8** | 218.7 | **124.4** | 251.4 |
| VED | **44570.4** | 52442.9 | 8755.4 | **8636.3** |

Table 2: Performance of our approach PS-GM versus the baseline (BL) for image completion. We report Fréchet distance for GAN-based models, and negative log-likelihood (NLL) for the VED.

## 4 EXPERIMENTS

We perform two experiments—one for generation from scratch and one for image completion. We find substantial improvement in both tasks. Details on neural network architectures are in Appendix A, and additional examples for image completion are in Appendix B.

### 4.1 DATASETS

**Synthetic dataset.** We developed a synthetic dataset based on MNIST. Each image consists of a $9 \times 9$ grid, where each grid cell is $16 \times 16$ pixels. Each grid cell is either filled with a colored MNIST digit or a solid color background. The program structure is a 2D repeating pattern of an MNIST digit; to add natural noise, we each iteration of the for-loop in a sketch $s_h$ renders different MNIST digits, but with the same MNIST label and color. Additionally, we chose the program structure to contain correlations characteristic of real-world images—e.g., correlations between different parts of the program, correlations between the program and the background, and noise in renderings of the same component. Examples are shown in Figure 4. We give details of how we constructed this dataset in Appendix A. This dataset contains 10,000 training and 500 test images.

**Facades dataset.** Our second dataset consists of 1855 images (1755 training, 100 testing) of building facades.[6] These images were all scaled to a size of $256 \times 256 \times 3$ pixels, and were divided into a grid of $15 \times 15$ cells each of size 17 or 18 pixels. These images contain repeating patterns of objects such as windows and doors.

### 4.2 GENERATION FROM SCRATCH

**Experimental setup.** First, we evaluate our approach PS-GM applied to generation from scratch. We focus on the synthetic dataset—we found that our facades dataset was too small to produce meaningful results. For the first stage of PS-GM (i.e., generating the program $P = (s, c)$), we use a LSTM architecture for the encoder $p_\phi(s, c \mid z)$ and a feedforward architecture for the decoder $q_{\tilde{\phi}}(z \mid s, c)$. As described in Section 2, we use Algorithm 1 to synthesize programs $P_x = (s_x, c_x)$ representing each training image $x \in X_{\text{train}}$. Then, we train $p_\phi$ and $q_{\tilde{\phi}}$ on the training set of programs $\{P_x \mid x \in X\}$.

For the second stage of PS-GM (i.e., completing the structure rendering $x_{\text{struct}}$ into an image $x$), we have applied a variational encoder-decoder (VED) architecture, in which the goal is to minimize the reconstruction error of the decoded image (trained on full images that had their representative program extracted). The encoder of the VAE architecture, $q_\theta(w \mid x_{\text{struct}})$, maps $x_{\text{struct}}$ to a latent vector $w$; this model has a standard convolutional architecture with 4 layers. The decoder $p_\theta(x \mid w)$ maps the latent vector to a whole image, and has a standard transpose convolutional architecture with 6 layers. We train $p_\theta$ and $q_\theta$ using the typical loss based on the VAE approach. We also trained a model based on Cycle-GAN which mapped renderings of programs to complete images. While it is difficult to compare objectively to a VAE architecture, it appeared to be at least as capable of

---

[6]We chose a large training set since our dataset is so small.

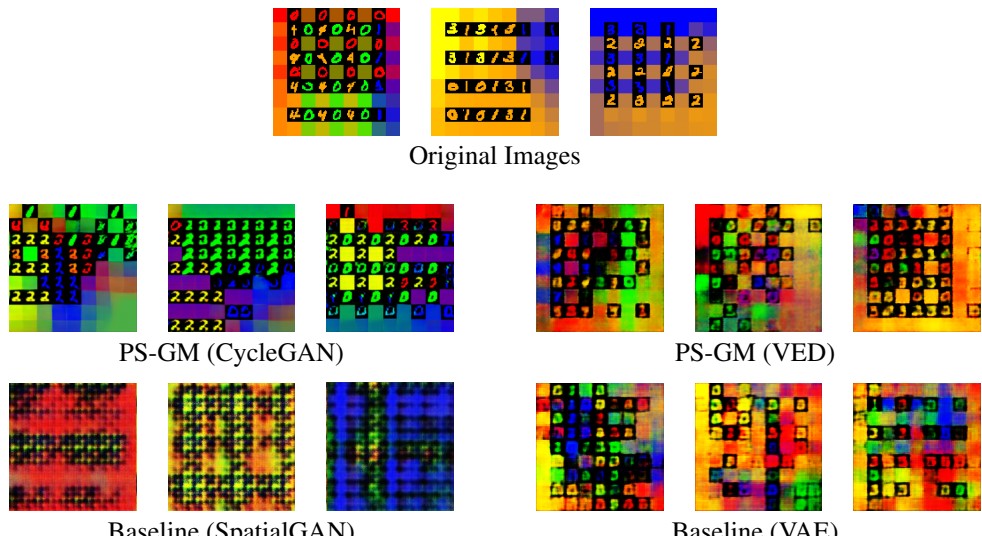

Original Images

PS-GM (CycleGAN)                    PS-GM (VED)

Baseline (SpatialGAN)              Baseline (VAE)

Figure 4: Examples of synthetic images generated using our approach, PS-GM (with VED and CycleGan), and the baseline (a VAE and a SpatialGAN). Images in different rows are unrelated since the task is generation from scratch.

generating structured images, and the structure in the PS-CycleGAN images were more apparent than in the PS-VAE due to less blurriness.

We compare our encoder-decoder approach to a baseline consisting of a vanilla VAE. The architecture of the vanilla VAE is the same as the architecture of the VED we used for the second stage of PS-GM, except the input to the encoder is the original training image $x$ instead of the structure rendering $x_{\text{struct}}$.

**Results.** We measure performance for PS-GM with the VED and the baseline VAE using the variational lower bound on the negative log-likelihood (NLL) Zhao et al. (2017) on a held-out test set. For our approach, we use the lower bound (2),[7] which is the sum of the NLLs of the first and second stages; we report these NLLs separately as well. Figure 4 in Appendix B shows examples of generated images. For PS-GM and SpatialGAN, we use Fréchet inception distance Heusel et al. (2017). Table 1 shows these metrics of both our approach and the baseline.

**Discussion.** The models based on our approach quantitatively improve over the respective baselines. The examples of images generated using our approach with VED completion appear to contain more structure than those generated using the baseline VAE. Similarly, the images generated using our approach with CycleGAN clearly capture more complex structure than the unbounded 2D repeating texture patterns captured by SpatialGAN.

### 4.3 IMAGE COMPLETION

**Experimental setup.** Second, we evaluated our approach PS-GM for image completion, on both our synthetic and the facades dataset. For this task, we compare using three image completion models: GLCIC IIzuka et al. (2017), CycleGAN Zhu et al. (2017), and the VED architecture described in Section 4.2. GLCIC is a state-of-the-art image completion model. CycleGAN is a generic image-to-image transformer. It uses unpaired training data, but we found that for our task, it outperforms approaches such as Pix2Pix Isola et al. (2017) that take paired training data. For each model, we trained two versions:

- **Our approach (PS-GM):** As described in Section 2 (for image completion), given a partial image $x_{\text{part}}$, we use Algorithm 1 to synthesize a program $P_{\text{part}}$. We extrapolate $P_{\text{part}}$ to

---

[7]Technically, $p_\theta(x \mid s_x, c_x)$ is lower bounded by the loss of the variational encoder-decoder.

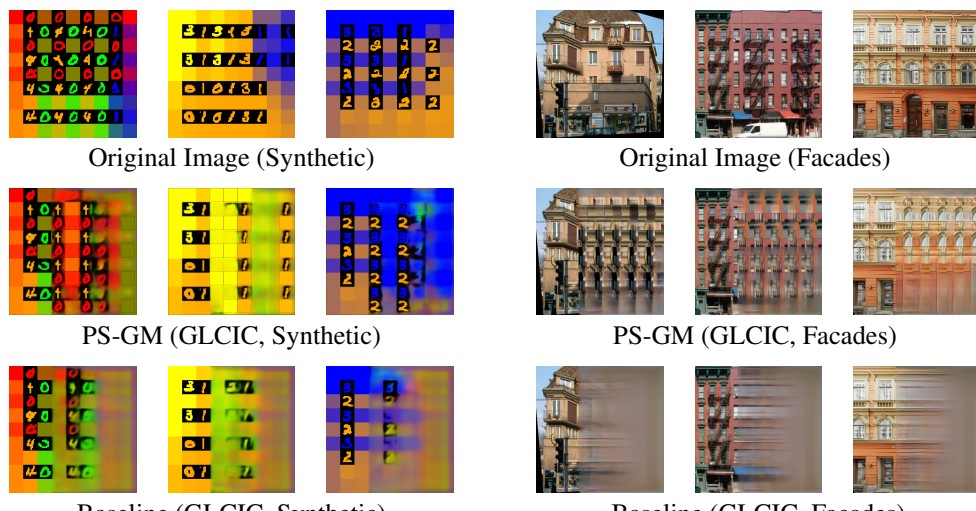

Figure 5: Examples of images generated using our approach (PS-GM) and the baseline, using GLCIC for image completion.

$\hat{P} = f(P_{\text{part}})$, and execute $\hat{P}$ to obtain a structure rendering $x_{\text{struct}}$. Finally, we train the image completion model (GLCIC, CycleGAN, or VED) to complete $x_{\text{struct}}$ to the original image $x^*$.

- **Baseline:** Given a partial image $x_{\text{part}}$, we train the image completion model (GLCIC, CycleGAN, or VED) to directly complete $x_{\text{part}}$ to the original image $x*$.

**Results.** As in Section 4.2, we measure performance using Fréchet inception distance for GLCIC and CycleGAN, and negative log-likelihood (NLL) to evaluate the VED, reported on a held-out test set. We show these results in Table 2. We show examples of completed image using GLCIC in Figure 5. We show additional examples of completed images including those completed using CycleGAN and VED in Appendix B.

**Discussion.** Our approach PS-GM outperforms the baseline in every case except the VED on the facades dataset. We believe the last result is since both VEDs failed to learn any meaningful structure (see Figure 7 in Appendix B).

A key reason why the baselines perform so poorly on the facades dataset is that the dataset is very small. Nevertheless, even on the synthetic dataset (which is fairly large), PS-GM substantially outperforms the baselines. Finally, generative models such as GLCIC are known to perform poorly far away from the edges of the provided partial image IIzuka et al. (2017). A benefit of our approach is that it provides the global context for a deep-learning based image completion model such as GLCIC to perform local completion.

## 5 CONCLUSION

We have proposed a new approach to generation that incorporates programmatic structure into state-of-the-art deep learning models. In our experiments, we have demonstrated the promise of our approach to improve generation of high-dimensional data with global structure that current state-of-the-art deep generative models have difficulty capturing. We leave a number of directions for future work. Most importantly, we have relied on a custom synthesis algorithm to eliminate the need for learning latent program structure. Learning to synthesize latent structure during training is an important direction for future work. In addition, future work will explore more expressive programmatic structures, including if-then-else statements.

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

## A  EXPERIMENTAL DETAILS

### A.1  SYNTHETIC DATASET

To sample a random image, we started with a $9 \times 9$ grid, where each grid cell is $16 \times 16$ pixels. We randomly sample a program $P = ((s_1, c_1), ..., (s_k, c_k))$ (for $k = 12$), where each perceptual component $c$ is a randomly selected MNIST image (downscaled to our grid cell size and colorized). To create correlations between different parts of $P$, we sample $(s_h, c_h)$ depending on $(s_1, c_1), ..., (s_{h-1}, c_{h-1})$. First, to sample each component $c_h$, we first sample latent properties of $c_h$ (i.e., its MNIST label $\{0, 1, ..., 4\}$ and its color $\{$red, blue, orange, green, yellow$\}$). Second, we sample the parameters of $s_h$ conditional on these properties. To each of the 25 possible latent properties of $c_h$, we associate a discrete distribution over latent properties for later elements in the sequence, as well as a mean and standard deviation for each of the parameters of the corresponding sketch $s_h$.

We then render $P$ by executing each $(s_h, c_h)$ in sequence. However, when executing $(s_h, c_h)$, on each iteration $(i, j)$ of the for-loop, instead of rendering the sub-image $c_h$ at each position in the grid, we randomly sample another MNIST image $c_h^{(i,j)}$ with the same label as $c_h$, recolor $c_h^{(i,j)}$ to be the same color as $c_h$, and render $c_h^{(i,j)}$. By doing so, we introduce noise into the programmatic structure.

### A.2  GENERATION FROM SCRATCH

**PS-GM architecture.**  For the first stage of PS-GM (i.e., generating the program $P = (s, c)$), we use a 3-layer LSTM encoder $p_\phi(s, c \mid z)$ and a feedforward decoder $q_{\tilde{\phi}}(z \mid s, c)$. The LSTM includes sequences of 13-dimensional vectors, of which 6 dimensions represent the structure of the for-loop being generated, and 7 dimensions are an encoding of the image to be rendered. The image compression was performed via a convolutional architecture with 2 convolutional layers for encoding and 3 deconvolutional layers for decoding.

For the second stage of PS-GM (i.e., completing the structure rendering $x_{\text{struct}}$ into an image $x$), we use a VED; the encoder $q_\theta(w \mid x_{\text{struct}})$ is a CNN with 4 layers, and the decoder $p_\theta(x \mid w)$ is a transpose CNN with 6 layers. The CycleGAN model has a discriminator with 3 convolutional layers and a generator which uses transfer learning by employing the pre-trained ResNet architecture.

**Baseline architecture.**  The architecture of the baseline is a vanilla VAE with the same as the architecture as the VED we used for the second state of PS-GM, except the input to the encoder is the original training image $x$ instead of the structure rendering $x_{\text{struct}}$. The baselines with CycleGAN also use the same architecture as PS-GM with CycleGAN/GLCIC. The Spatial GAN was trained with 5 layers each in the generative/discriminative layer, and 60-dimensional global and 3-dimensional periodic latent vectors.

### A.3  IMAGE COMPLETION.

**PS-GM architecture.**  For the first stage of PS-GM for completion (extrapolation of the program from a partial image to a full image), we use a feedforward network with three layers. For the second stage of completion via VAE, we use a convolutional/deconvolutional architecture. The encoder is a CNN with 4 layers, and the decoder is a transpose CNN with 6 layers. As was the case in generation, the CycleGAN model has a discriminator with 3 convolutional layers and a generator which uses transfer learning by employing the pre-trained ResNet architecture.

**Baseline architecture.**  For the baseline VAE architecture, we used a similar architecture to the PS-GM completion step (4 convolutional and 6 deconvolutional layers). The only difference was the input, which was a partial image rather than an image rendered with structure. The CycleGAN architecture was similar to that used in PS-GM (although it mapped partial images to full images rather than partial images with structure to full images).

# B  ADDITIONAL RESULTS

In Figure 6, we show examples of how our image completion pipeline is applied to the facades dataset, and in Figure 7, we show examples of how our image completion pipeline is applied to our synthetic dataset.

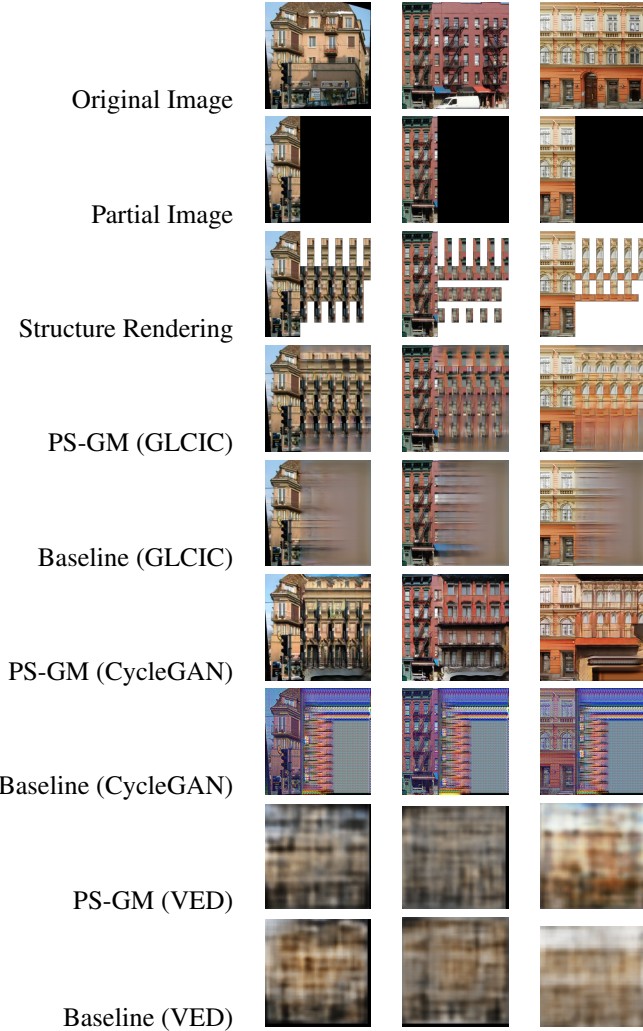

Figure 6: Examples of our image completion pipeline on the facades dataset.

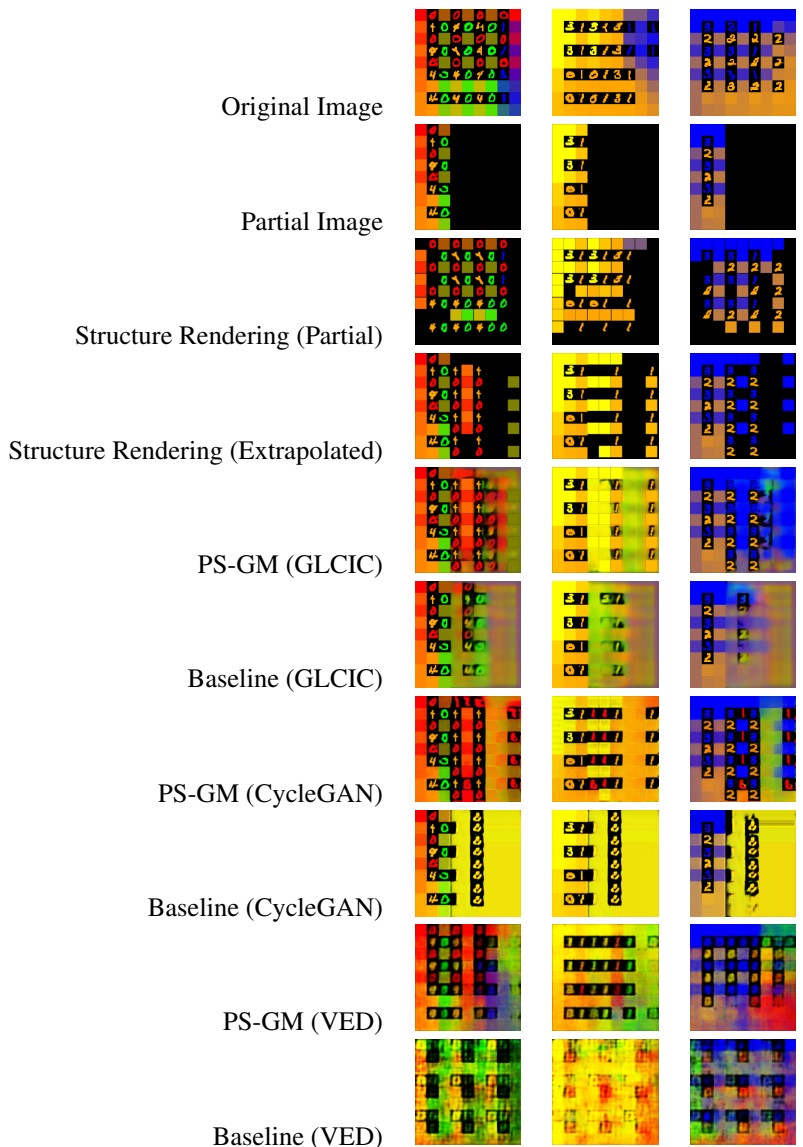

Original Image

Partial Image

Structure Rendering (Partial)

Structure Rendering (Extrapolated)

PS-GM (GLCIC)

Baseline (GLCIC)

PS-GM (CycleGAN)

Baseline (CycleGAN)

PS-GM (VED)

Baseline (VED)

Figure 7: Examples of our image completion pipeline on our synthetic dataset.

