# OpenReview forum: "LEARNING NEUROSYMBOLIC GENERATIVE MODELS VIA PROGRAM SYNTHESIS"
_ICLR.cc/2019/Workshop/drlStructPred — drlStructPred 2019_

### Official Review · AnonReviewer3 · 2019-04-04
**The paper presents an program synthesis method for image generation, experiments focus on domain specific/synthetic datasets where the models are expected to work well, but does have potential to be applied on more general datasets**

**Rating:** 4
**Confidence:** 3

**Review:**

The paper presents an image generation model, focusing on learning a program that explores the regularities within the data, such as cycling patterns. The model aims at learning two hidden variables, one that defines the structure of the program and another that defines how the model fills the patches defined within the program.

I believe this is a good direction for research in image generation, or any generation tasks of similar nature. Such models would allow for computationally efficient models, especially for generating higher resolution images, since there is a quadratic increase of the number of parameters needed. It would be great that in the future versions of the paper the authors could show that larger images can be learned with similar numbers of model parameters.

Finally, I would like to see results on medium to larger scale datasets, such as CIFAR-10 or Imagenet. Not only samples are more diversified, it would also allow for an analysis of the sample efficiency of the proposed method compared to existing methods.

---

### Official Review · AnonReviewer4 · 2019-04-05
**Interesting technique to incorporate programs representing global structures in generative models**

**Rating:** 3
**Confidence:** 2

**Review:**

This paper presents a technique to incorporate programs (representing global structure in images) into the generative model thereby capturing more global structure in image generation and image completion tasks. For image generation tasks, it first samples a latent vector z that is used to then sample a program (s,c). The sampled program is then executed to generate x_struct, which is afterwards used to generate the image x using image completion techniques such as GLCIC and CycleGAN. For image completion tasks, it first samples a partial program P_part from a partial image x_part, and then extrapolates the partial program to generate a full program P. The full program is generated to obtain x_part+struct, which is then completed to obtain the full image. The approach is evaluated on synthetic and a real-world dataset of building facades, and it outperforms several baseline models.

Overall, I really liked the idea of introducing programs as structural biases in the generative models, where the programs comprise of both symbolic (for loops) and neural (sub-images) components. While there are some recent approaches that have looked at learning such neuro-symbolic programs, this seems to be one of the first approaches for learning generative models of images. The idea of conditioning the image completion models on output obtained by evaluating such intermediate programs and extrapolating programs for completion tasks is also quite nice.

Although this is an interesting first step, many of the design choices seem a bit restrictive for more general image generation tasks. For example, the paper only considers one for-loop template:
for (i,j):
  draw (a*i+b, a'*i+b', ??)
It might be interesting to extend the class of templates to include nested loops with conditionals to capture more general global structures together with corner cases.

The search algorithm for synthesizing programs performs an exhaustive search and it is not clear if such an enumerative strategy would scale to richer program templates.

For the VAE decoder, is it the case that the sequence decoder decodes the for loop structure program directly?

It would also be good to provide a bit more details about the program extrapolation part for image completion model. For example, is there an assumption that all images are of equal size for the extrapolation model or is the model conditioned on the final expected size of the image?

---

### Official Review · AnonReviewer1 · 2019-04-06
**Interesting task.**

**Rating:** 3
**Confidence:** 2

**Review:**

This work consider the problem of generating or completing images with repeated structures. The main idea is to have a generative model which generates image x from hidden programs P, which is in turn generated from hidden embedding z.

The proposed model is mainly a VAE of programs P extended with components for the image x -- 1) domain greedy heuristics to generate P from x, and 1) CycleGAN to generate x from x_structure which is generated from P in a deterministic fashion.

I find the task very interesting with potentially practical applications. The result on synthetic data looks pretty good (much better than VAE and SpatialGAN). The result on real dataset (1855 Facades) still need to be improved. It could be that the data set is just too small for the proposed model. It could also be search failure when generating P from x. Or there might be other reasons?

---

### Official Review · AnonReviewer5 · 2019-04-06
**Novel approach to improving VAEs**

**Rating:** 3
**Confidence:** 2

**Review:**

This paper is written clearly and is well thought out in its approach and results. Their contributions are clear, although ambitious as they aim to improve image reconstruction by providing a VAE a reconstructed image based on a grid image created via program synthesis. Their work does show improvement over most state of the art based on the Frechet distance as a performance measure in both their synthetic baseline as well as the Facades baseline other than VED which outperforms the proposed model. It can also be argued though that their improvements may be contributed to the augmented input overfitting the original structure. For samples in their work in which they are compared to VED, it is clear that their work does better represent the original structure, but it is not clear if their method has created a faithful reconstruction of the image or simply  a grid of an average part of the inputs which overfit to the given input. Overall, I believe this work is interesting and may be moving in the right direction for improving image reconstruction algorithms although it is an early start.

---

### Decision · Program_Chairs · 2019-04-08
**Acceptance Decision**

Accept